**Data Availability Statement:** All relevant data are within the paper and its Supporting Information files.

# A prospective diagnostic evaluation of accuracy of self-taken and healthcare worker-taken swabs for rapid COVID-19 testing

**Helen R. Savage**[1], **Lorna Finch**[2], **Richard Body**[3], **Rachel L. Watkins**[2], **LSTM Diagnostics group**[2¶], **CONDOR steering group**[3¶], **Gail Hayward**[4], **Eloïse Cook**[3], **Ana I. Cubas-Atienzar**[2], **Luis E. Cuevas**[1], **Peter MacPherson**[1,5,6], **Emily R. Adams**[2]*

1 Department of Clinical Sciences, Liverpool School of Tropical Medicine and Hygiene, Liverpool, United Kingdom, 2 Centre for Drugs and Diagnostics, Liverpool School of Tropical Medicine and Hygiene, Liverpool, United Kingdom, 3 Manchester University NHS Foundation Trust, Research and Innovation, Manchester, United Kingdom, 4 Nuffield Department of Primary Care Health Sciences, University of Oxford, Oxford, United Kingdom, 5 Malawi-Liverpool-Wellcome Trust Clinical Research Programme, Blantyre, Malawi, 6 Clinical Research Department, London School of Hygiene and Tropical Medicine, London, United Kingdom

¶ Membership of the LSTM Diagnostics group and CONDOR steering group is provided in the Acknowledgments.
* Emily.adams@lstmed.ac.uk

## Abstract

### Background

Rapid diagnostic tests (RDTs) developed for point of care detection of SARS-CoV-2 antigen are recommended by WHO to use trained health care workers to collect samples. We hypothesised that self-taken samples are non-inferior for use with RDTs to diagnose COVID-19. We designed a prospective diagnostic evaluation comparing self-taken and healthcare worker (HCW)-taken throat/nasal swabs to perform RDTs for SARS-CoV-2, and how these compare to RT-PCR.

### Methods

Eligible participants 18 years or older with symptoms of COVID-19. 250 participants recruited at the NHS Test and Trace drive-through community PCR testing site (Liverpool, UK); one withdrew before analysis. Self-administered throat/nasal swab for the Covios® RDT, a trained HCW taken throat/nasal sample for PCR and HCW comparison throat/nasal swab for RDT were collected. RDT results were compared to RT-PCR, as the reference standard, to calculate sensitivity and specificity.

### Findings

Seventy-five participants (75/249, 30.1%) were positive by RT-PCR. RDTs with self-taken swabs had a sensitivity of 90.5% (67/74, 95% CI: 83.9–97.2), compared to 78.4% (58/74, 95% CI: 69.0–87.8) for HCW-taken swabs (absolute difference 12.2%, 95% CI: 4.7–19.6, p = 0.003). Specificity for self-taken swabs was 99.4% (173/174, 95% CI: 98.3–100.0), versus

**Funding:** All authors have completed the ICMJE uniform disclosure form at http://www.icmje.org/disclosure-of-interest/ and declare: This study received funding from the UK Research Council through a PhD scholarship from the MRC Doctoral Training Partnership to HRS. PM is funded by Wellcome (200901/Z/16/Z), Wellcome Trust award. 'Detecting and Excluding Coronavirus disease 2019 (COVID-19) at the Point of Need' (220764/Z/20/Z), FALCON C-19 study was funded by National Institute for Health Research (NIHR), Asthma UK and the British Lung Foundation; and is supported by the global alliance for diagnostics (FIND). ERA contributed to this study design and analysis in her role as PhD supervisor to HRS at LSTM, she is also Director of Epidemics and NTDs at Mologic Ltd a UK diagnostics company who provided the RDTs for this study under joint Wellcome funding. This does not alter our adherence to PLOS ONE policies on sharing data and materials. For the purpose of open access, the author has applied a CC BY public copyright licence to any Author Accepted Manuscript version arising from this submission. The funders had no role in study design, data collection and analysis, decision to publish, or preparation of the manuscript.

**Competing interests:** ERA contributed to this study design and analysis in her role as PhD supervisor to HRS at LSTM, she is also Director of Epidemics and NTDs at Mologic Ltd a UK diagnostics company who provided the RDTs for this study under joint Wellcome funding. This does not alter our adherence to PLOS ONE policies on sharing data and materials.

98.9% (172/174, 95% CI: 97.3–100.0) for HCW-taken swabs (absolute difference 0.6%, 95% CI: 0.5–1.7, p = 0.317). The PPV of self-taken RDTs (98.5%, 67/68, 95% CI: 95.7–100.0) and HCW-taken RDTs (96.7%, 58/60, 95% CI 92.1–100.0) were not significantly different (p = 0.262). However, the NPV of self-taken swab RDTs was significantly higher (96.1%, 173/180, 95% CI: 93.2–98.9) than HCW-taken RDTs (91.5%, 172/188, 95% CI 87.5–95.5, p = 0.003).

## Interpretation

In conclusion, self-taken swabs for COVID-19 testing offer an accurate alternative to healthcare worker taken swabs for use with RDTs. Our results demonstrate that, with no training, self-taken throat/nasal samples can be used by lay individuals as part of rapid testing programmes for symptomatic adults. This is especially important where the lack of trained healthcare workers restricts access to testing.

## Introduction

The Severe Acute Coronavirus 2 (SARS-CoV-2) is a novel pathogen causing Coronavirus Disease-19 (COVID-19) that emerged in December 2019 and spread quickly around the globe before being declared a pandemic on 20th March 2020. Confirmation of SARS-CoV-2 infection is recommended by real-time polymerase chain reaction (RT-PCR) testing, however this requires well-resourced laboratory facilities, which are not available in many settings [1]. Given the need to rapidly upscale testing, rapid diagnostic tests (RDTs) were developed to detect SARS-CoV-2 antigen(s) (Ag), which can be used at point of care without a laboratory infrastructure. Guidance from the World Health Organization (WHO) recommends using RDTs in settings with trained health workers to facilitate collecting samples and processing tests [1]. Currently, large-scale self-testing for SARS-CoV-2 is conducted in schools, workplaces, and homes in the U.K.; however concerns have been raised over the accuracy of these tests and the risk of missing infected individuals [2]. Many tests designed and regulated for "Professional Use Only" have been implemented for self-testing use, but little accuracy information exists for self-swabbing and interpretation.

Previous work has suggested that RDTs achieve higher sensitivity when performed by laboratory scientists (sensitivity of the Innova lateral flow test 79%, 95% CI 72–84%) than by healthcare workers (sensitivity 70%, 95% CI 63–76%) [3]. Sensitivity can be substantially affected by the quality of the sample and swabbing technique [4]. A small number of studies have compared self-taken to healthcare worker (HCW) taken swabs for RT-PCR, with a high degree of concordance [5, 6]. To the best of our knowledge, no studies have compared self-taken and HCW-taken samples with identical swab types and identical RDTs rather than comparing alternative sampling strategies.

We therefore set out to compare the sensitivity and specificity of self-taken and HCW-taken throat/nasal swabs to perform a RDT for SARS-CoV-2, and how these sampling approaches perform compared to the RT-PCR. We hypothesised that, if self-taken samples are accurate for use with RDTs in clinical and research settings, this could have substantial individual and public health benefit.

## Methods

We conducted a prospective diagnostic accuracy evaluation to compare self-taken and HCW-taken throat/nasal swabs RDTs with a standard HCW-taken throat/nasal swab tested using RT-PCR. Participants were recruited as part of the 'Facilitating Accelerated Clinical Evaluation of Novel Diagnostic Tests for COVID -19 (FALCON), workstream C (undifferentiated community testing)' [7], which aims to evaluate the diagnostic accuracy of commercially supplied in-vitro diagnostic (IVD) tests for SARS-CoV-2 infection.

Participants were recruited consecutively when presenting at the Liverpool John Lennon Airport (Liverpool, UK) drive through community PCR testing site, a National Health Service (NHS) Test and Trace site for the general population with symptoms of COVID-19, defined as a high temperature, continuous cough or change in sense of taste or smell. People presenting for testing were assessed for eligibility in their vehicle and received a patient information sheet prior to offer of NHS test (which was taken after the study). Eligible participants were 18 years or older who verbally confirmed they currently had symptoms of COVID-19. If multiple occupants were in the vehicle, they were each assessed for eligibility. Informed verbal consent was taken and recorded by a researcher on site. Demographic and self-reported symptom data were recorded electronically; a list of symptoms from the FIND alliance and participants could also list additional symptoms they felt were relevant. We excluded anyone under 18 years, or who did not state they currently had COVID-19 symptoms, or if they did not consent to participate. We did not record people who declined to participate. As people presented to the testing centre in vehicles, we were unable to distinguish between drivers or relatives, and people presenting for testing services.

A HCW in personal protective equipment (PPE, in line with local guidelines) passed a self-collection kit for each participant into the vehicle. Each kit contained a short instruction sheet taken from the manufacturer protocol (see S1 File), a tissue, a swab, and collection tube. The participant self-administered their throat/nasal swab without further advice, observation, or supervision. Once complete the participant signaled the research team and a trained healthcare worker took a throat/nasal sample using a COPAN mini UTM (universal transport medium) kit 1ml for PCR from one side and a comparison swab for RDT from the other. All swabs taken for RDT were randomly numbered so that laboratory staff performing the RDTs were blinded to sample collection method and could not identify paired samples.

Samples were transported to the Liverpool School of Tropical Medicine (LSTM) by trained research staff (in accordance with the requirements for Category B substance UN3373 [8]) where they were processed and tested in a category 3 laboratory within 3 hours of sampling. UTM samples for PCR were aliquoted and frozen at -80˚C. The Covios® COVID-19 Antigen Rapid Diagnostic test, which detects the SARS-CoV-2 nucleoprotein, was used for testing all RDT samples [9]. This test is CE marked and manufactured in the UK by Global Access Diagnostics (legal manufacturer Mologic) (patients 1–100 LOT: CALFD-102-1, patient 101–250 LOT: CALFD-130-1). RDTs were run according to the manufacturer's instructions, this includes grading the result line from 0–10 on the RDT using a visual reference card. Each test was read by two trained researchers, if there was a disagreement between the two readers, a third reader was requested.

RNA was extracted from batched UTM samples using the QiAamp96 Virus Qiacube HT kit and RT-PCRs were run following manufacturer's instruction using TaqPath COVID-19 RT-PCR on QuantStudio 5 (ThermoFisher). RT-PCR reactions volumes were made in 20 µl. Reverse transcription step was performed at 53˚C for 10 minutes and this was followed by an activation step of 2 minutes at 95˚C, then PCR was carried for 40 amplification cycles at 95˚C for 3 minutes and 60˚C for 30 seconds. Fluoresce was recorded in the FAM, VIC, ABY and

JUN channels for the ORF1ab, N, S and MS2 targets respectively. RT-PCR was used as the reference standard test in this study for comparison of the RDT results.

## Sample size and statistical analysis

Sample size was calculated using an alpha of 0.05, anticipated prevalence of 20% (based on the positivity rate of PCR tests of individuals presenting for testing within Liverpool in the week commencing 21st January 2021, calculated by Public Health England [10]), minimum test sensitivity of 80%, specificity 99%, and precision interval of 10% [11]. These gave a planned sample size of 308. We described participant characteristics using summary statistics and compared self-swab sampling RDT results with RT-PCR results. RDTs were graded 0–10, with 0 representing a negative result, and 1–10 positive results based on a visual scale using the manufacturer's reference card. For each RT-PCR result with three target genes a mean of the three CT values was taken to give a single RT-PCR CT result for each sample; RT-PCR results with mean cycle threshold (CT) values <40 were considered positive and CT values ≥40 were graded as negative. We calculated sensitivity and specificity, and positive and negative predictive values (PPV and NPV), all with binomial exact 95% confidence intervals (CI) in R v4.1.1 (R Foundation for Statistical Computing, Vienna). Paired results were compared between self-taken and HCW-taken samples using McNemar's test. Indeterminate RDT results were recorded but excluded from further analysis. Indeterminate RT-PCR results were repeated twice, and the repeat test result was used for analysis. RT-PCR data were classified according to the mean RT-PCR CT threshold values (<20, 20–24.9, 25–29.9, 30–34.9 and ≥35).

Recruitment started on 31st March 2021 and 100 participants had been recruited until the 21st May 2021, when local COVID-19 prevalence declined (positivity testing rate 0.4%) giving very small numbers of positive cases. Recruitment was temporarily halted until July 2021 when prevalence increased (positivity testing rate 12.9%) and a further 150 participants were recruited. Recruitment ended on 9th August 2021.

## Ethical approval

Ethical approval was obtained from the National Research Ethics Service (reference 20/WA/0169) and the Health Research Authority (IRAS ID:28422, clinical trial ID: NCT04408170).

## Role of the funding source

The funders had no role in study design, data collection and analysis, decision to publish, or preparation of the manuscript.

## Results

Two hundred and fifty participants were recruited between the 31st March 2021 and the 9th August 2021. One participant withdrew after recruitment and did not wish data or samples to be included, leaving 249 participants for the analysis. The mean age of participants was 40 years (range 18–82, interquartile range [IQR] 30.0–50.0 years), 104 (41.7%) were male and 216 white British (86.7%) (Table 1). One hundred and eighty (72.3%) had received at least one vaccine dose against SARS-CoV-2 and of these 113 had received a second dose. The time interval since vaccination and the vaccine brand were not available. The most common self-reported symptoms by participants were cough (174, 69.9%), fever (78, 31.3%), sore throat (71, 28.5%) and headache (53, 21.3%); all participants were symptomatic. The median duration since symptom onset was 2 days (IQR 1–3 days). A full data table of all participant characteristics and results is available in the supplementary data.

**Table 1. Characteristics of participants.**

|  | All (N, %) | RT-PCR positive (N, %) |
|---|---|---|
| All | 249 | 75 |
| Age in years, mean, (range, IQR) | 40 (18–82, 30.0–50.0) | 37.6 (18–70, 24.5–50.0) |
| Male | 104 (41.7) | 42 (56.0) |
| Median symptom duration (days), range, IQR | 2.0 (0–33, 1–3) | 2.0 (0–32, 1–3) |
| Shortness of breath | 7 (2.8) | 2 (2.7) |
| Cough | 174 (69.9) | 53 (70.7) |
| Fever | 78 (31.3) | 31 (41.3) |
| Chest pain | 11 (4.4) | 7 (9.3) |
| Sore throat | 71 (28.5) | 19 (25.3) |
| Confusion | 0 (0) | 0 (0) |
| Rash | 1 (0.4) | 1 (1.3) |
| Loss of smell | 27 (10.8) | 17 (22.7) |
| Loss of taste | 25 (10.0) | 16 (21.3) |
| Abdominal pain | 6 (2.4) | 1 (1.3) |
| Vomiting | 7 (2.8) | 3 (4.0) |
| Diarrhoea | 11 (4.4) | 3 (4.0) |
| Headache | 53 (21.3) | 22 (29.3) |
| Tiredness/Fatigue | 10 (4.0) | 5 (6.7) |
| Tight chest | 1 (0.4) | 0 (0) |
| Other | 70 (28.1) | 31 (41.3) |
| White British | 216 (86.7) | 67 (89.3) |
| Irish | 10 (4.0) | 2 (2.7) |
| Other white | 7 (2.8) | 1 (1.3) |
| Indian | 2 (0.8) | 0 (0.0) |
| Mixed ethnic group | 9 (3.6) | 3 (4.0) |
| Other ethnic group | 5 (2.0) | 2 (2.7) |
| Vaccinated 1st dose | 180 (72.3) | 52 (69.3) |
| Vaccinated 2nd dose | 113 (45.4) | 32 (42.7) |

Seventy-five participants (75/249, 30.1%) tested positive by RT-PCR. The mean age of RT-PCR positive participants was 37.6 years (range 18–70, IQR 24.5–50.0), 42 (56.0%) were male and 67 were white British (89.3%) (Table 1). Fifty-two (69.3%) of the 75 RT-PCR-positive participants had received a first COVID-19 vaccine dose and 32 (42.7%) a second dose. Since symptom onset, the median duration in days was 2 (IQR: 1–3) and the most commonly reported symptoms were cough (53, 70.3%), fever (31, 41.3%), headache (22, 29.3%), sore throat (19, 25.3%), loss of smell (17, 22.7%) and loss of taste (16, 21.3%).

Overall, self-taken throat/nasal RDTs were positive in 68/249 (27.3%, 95% CI: 21.9–33.3) participants, one (0.4%) was indeterminate and 180 (72.3%) negative. HCW-taken throat/nasal RTDs were positive in 61/249 (24.5%, 95% CI: 19.3–30.3) participants, none was indeterminate and 188 (75.5%) were negative. The participant with the indeterminate RDT was excluded from further analysis.

RDT kits using a self-taken swab had a sensitivity of 90.5% (67/74, 95% CI: 83.9–97.2) and specificity of 99.4% (173/174, 95% CI: 98.3–100.0) when compared to the reference standard (Table 2). HCW-taken RDTs had a sensitivity of 78.4% (58/74, 95% CI 69.0–87.8) and specificity of 98.9% (172/174, 95% CI: 97.3–100.0) compared to the reference standard. The difference in sensitivity was 12.2% (95% CI: 4.7–19.6, p = 0.003), the difference in specificity was 0.6% (95% CI: 0.5–1.7, p = 0.317). Of the self-HCW RDT pairs, 238/248 (96.0%) agreed, and 10/248

**Table 2. Sensitivity and specificity of self- and healthcare worker-taken swabs for COVID-19 rapid diagnostic testing.**

|  | Comparison to RT-PCR | | | | | | | |
|---|---|---|---|---|---|---|---|---|
|  | Sensitivity (%) | 95% CI | Specificity (%) | 95% CI | PPV (%) | 95% CI | NPV (%) | 95% CI |
| **Self-taken RDT** | 90.5 | 83.9–97.2 | 99.4 | 98.3–100.0 | 98.5 | 95.7–100.0 | 96.1 | 93.3–98.9 |
| **HCW RDT** | 78.4 | 69.0–87.8 | 98.9 | 97.3–100.0 | 96.7 | 92.1–100.0 | 91.5 | 87.5–95.5 |

* CI = Confidence intervals, PPV = Positive predictive value, NPV = Negative predictive value

(4.0%) were discordant (Fig 1). Of the discordant pairs, on nine occasions the self-taken swab RDT was read as positive, while the HCW-taken swab was read negative (Table 3). Nine of these pairs were RT-PCR positive and one negative. The discordant pair that was RT-PCR negative, the HCW-taken swab RDT was positive, while the self-taken RDT was negative. The PPV of self-taken RDTs (98.5%, 67/68, 95% CI: 95.7–100.0) and HCW-taken RDTs (96.7%, 58/60, 95% CI 92.1–100.0) were not significantly different (p = 0.262). However, the NPV of self-taken swab RDTs was significantly higher (96.1%, 173/180, 95% CI: 93.2–98.9) than HCW-taken RDTs (91.5%, 172/188, 95% CI 87.5–95.5, p = 0.003).

Sensitivity of the RDTs varied by mean CT values (Table 4 and Fig 2), a full table of mean and CT values from each gene tested is available in the supplementary data. Self-taken and HCW-taken samples with CT values <20 had 100% (32/32, CI: 89.1–100.0) sensitivity; samples with CT values 20–24.9 had 91.7% (22/24, 95% CI: 73.0–99.0) for self-taken and 83.3% (20/24, 95% CI: 62.6–95.3) for HCW-taken RDTs. At CT values between 25 and 29.9, RDTs sensitivity was 80.0% (12/15, 95% CI: 51.9–95.7) for self-taken and 40.0% (6/15, 95% CI: 16.3–67.7) for HCW-taken swabs, while at CT values 30–34.9 both self-taken and HCW-taken swabs had sensitivity of 33.0% (1/3, 95% CI: 0.8–90.6). Sensitivity for samples with CT values ≥35 was 0% (0/1, 95% CI: 0.0–97.5).

## Discussion

This study found that the sensitivity of self-taken swabs for the detection of SARS-CoV-2 antigen was higher (90.5%) than using HCW-taken swabs (78.3%), with similar specificity. No RT-PCR-positive results from HCW- taken swabs were missed by self-taken swabs and the PPV and NPV for both methods were over 90%.

Current WHO guidance for implementing RDTs indicates swabbing to collect samples should be conducted by trained professionals [1]. Having HCW take swabs requires training, PPE to be available, regular donning and doffing, and close contact with potentially infectious individuals. All of which pose expenditure and added risks to those performing sampling, especially in countries where the majority of HCW remain unvaccinated. Using self-sampling for testing could reduce the workload of health workers and increase the ability of services to test patients in both clinical and research settings where trained workers are not available. These results show self-taken throat-nasal samples with only written and pictorial instructions can be used by the general public for RDTs and is not likely to reduce the sensitivity of testing, which could widen access.

Within this study, self-taken swabs had higher sensitivity than HCW swabs for RDTs in a general population setting in the UK. The self-swabbing technique was not monitored for quality, no participants failed to take the swabs and no assistance was given so the results could be extrapolated to other non-supervised settings. The high concordance of self- and HCW-taken results has been reported from studies comparing self-and HCW-taken swabs for PCR testing and also within studies looking at alternative swab types (nasopharyngeal, nasal only) for RDTs [6, 12–14].

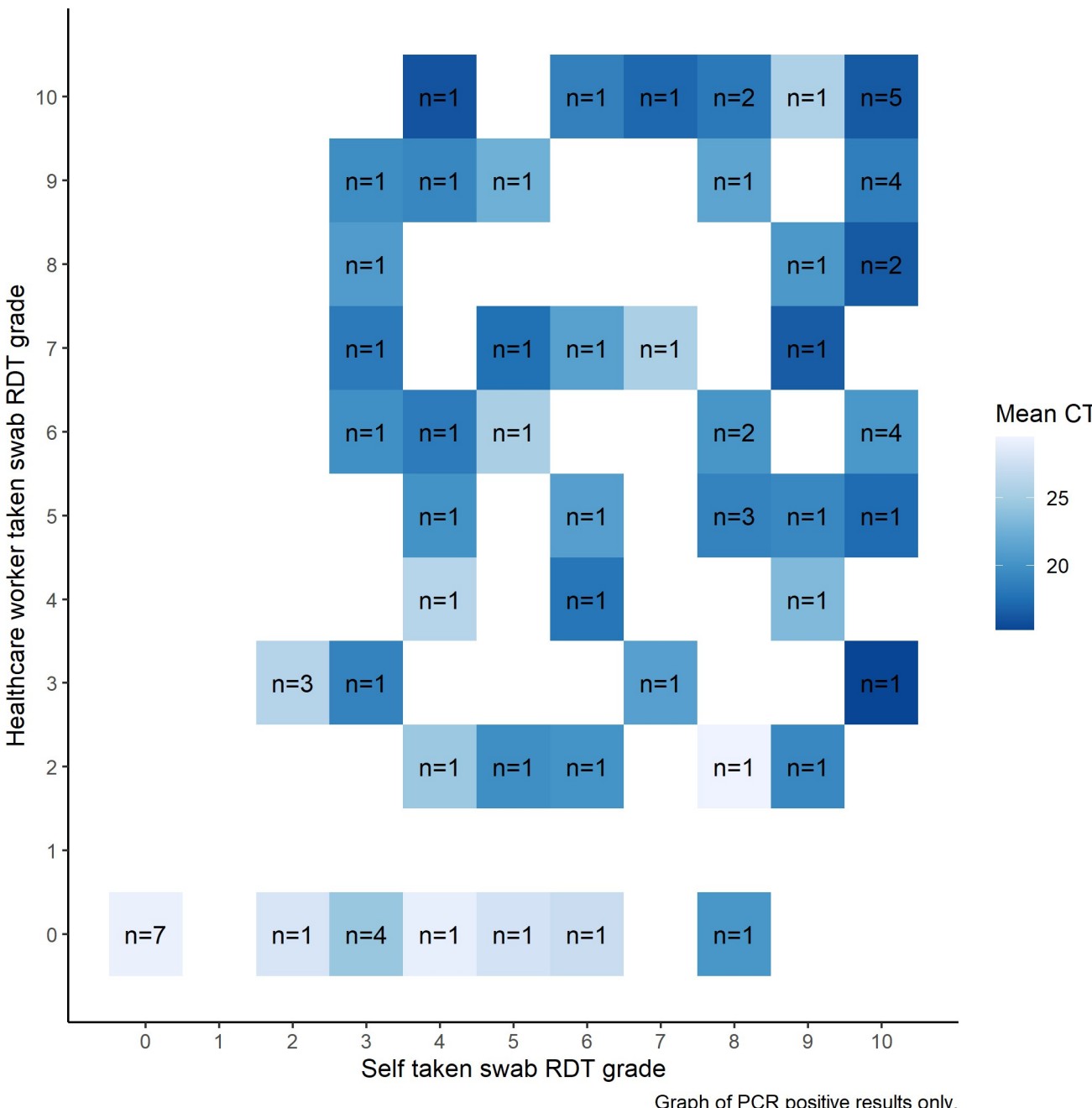

**Fig 1. Correlation between self and healthcare worker graded RDT result by PCR cycle threshold.**

The limitations of this study are that the sampling order was not randomized as the HCW swab for RDT was taken after the swab for the RT-PCR, and only from one nostril; this may lower sensitivity, or participants may experience sampling fatigue. However, previous studies have shown that repeated sampling from one nostril using nasal mid-turbinate specimens does not impact RT-PCR sensitivity or CT values so, although we used throat/nasal sampling, this repeated sampling may not be the reason for lower sensitivity [15]. Participation was voluntary so people who were less confident to take their own sample may not have agreed to

**Table 3. Table showing full results for participants with discrepant RDT results.**

| Participant | RT-PCR result (mean Ct value) | Self-taken RDT result (Reader 1/Reader 2) | HCW-taken RDT result (Reader 1/Reader 2) |
|---|---|---|---|
| 26 | Positive (29.03) | Positive (4/4) | Negative (0/0) |
| 38 | Negative (>40) | Negative (0/0) | Positive (2/2) |
| 106 | Positive (21.68) | Positive (3/3) | Negative (0/0) |
| 131 | Positive (27.99) | Positive (3/3) | Negative (0/0) |
| 140 | Positive (25.44) | Positive (3/3) | Negative (0/0) |
| 169 | Positive (28.09) | Positive (2/2) | Negative (0/0) |
| 189 | Positive (20.35) | Positive (8/8) | Negative (0/0) |
| 195 | Positive (22.96) | Positive (3/3) | Negative (0/0) |
| 231 | Positive (27.33) | Positive (6/6) | Negative (0/0) |
| 249 | Positive (27.86) | Positive (5/5) | Negative (0/0) |

take part; it is also likely participants may have done previous COVID-19 tests, and so have experience of self-sampling.

The majority of participants with discrepant results between self-taken and HCW taken RDT results had a mean CT of between 25–30 on PCR (six of ten). Although there was only a small number of participants with a CT value in this range, these values are influential in estimates of the sensitivity of the HCW taken RDT tests. Because of the small numbers, we did not undertake further formal statistical analysis; however when participants with CT values within this range were excluded, our main conclusion that self-taken swabs were non-inferior to healthcare worker taken swabs did not change (sensitivity with CT 25–29.9 excluded: self taken RDT 91.7% [55/60]; HCW taken 88.3% [53/55]). Similar results were noted in the trial by García-Fiñana et al. [4] who also saw a decrease in sensitivity of RDTs compared to PCR in samples with a CT of greater than 24.4. Future research could compare self-taken and healthcare worker-taken samples by PCR including housekeeper genes to see if the quality of sampling varies and may affect CT values, particularly for samples with CT values over 25, as lower

**Table 4. Sensitivity of self- and healthcare worker-taken swab for rapid diagnostic testing by RT-PCR CT ranges.**

| | RT-PCR CT range | | | | |
|---|---|---|---|---|---|
| | <20 | 20–24.9 | 25–29.9 | 30–34.9 | ≥35 |
| **Self-taken RDT** | | | | | |
| Positive | 32 | 22 | 12 | 1 | 0 |
| Negative | 0 | 1 | 3 | 2 | 1 |
| Indeterminate | 0 | 1 | 0 | 0 | 0 |
| Sensitivity | 100.0% | 91.7% | 80.0% | 33.3% | 0% |
| 95% CI | 89.1–100.0% | 73.0–99.0% | 51.9–95.7% | 0.8–90.6% | 0.0–97.5% |
| Cumulative sensitivity | 100.0% | 96.4% | 93.0% | 90.5% | 89.3% |
| 95% CI | 89.1–100.0% | 87.7–99.6% | 84.3–97.7% | 81.5–96.1% | 80.1–95.3% |
| **HCW RDT** | | | | | |
| Positive | 32 | 20 | 6 | 1 | 0 |
| Negative | 0 | 4 | 9 | 2 | 1 |
| Indeterminate | 0 | 0 | 0 | 0 | 0 |
| Sensitivity | 100% | 83.3% | 40.0% | 33.3% | 0.0% |
| 95% CI | 89.1–100.0% | 62.6–95.3% | 16.3–67.7% | 0.8–90.6% | 0.0–97.5% |
| Cumulative sensitivity | 100.0% | 92.9% | 81.7% | 79.7% | 78.7% |
| 95% CI | 89.1–100.0% | 82.7–98.0% | 70.7–89.9% | 68.8–88.2% | 67.6–87.3% |

sampling quality could result in lower CT values and therefore a perceived poorer performance in this range.

By focusing on differences in the sampling process (thus removing issues of running and interpretation of RDTs), we have shown that the ability of individuals to take their own samples is unlikely to explain the differences in test accuracy, and that if individuals self-take samples for RDTs, the results can be as accurate as professionally taken swabs. Previously in the UK RDT positive tests were confirmed with follow-on PCR tests which, given the high prevalence of circulating infection, and the high workloads led to laboratory errors. Recent test discrepancies in South-West England, which reported large numbers of RDT-positive but PCR-negative results, were due to incorrect PCR results, with correct RDTs results [16]. Our study suggest that the public and healthcare professionals should trust RDT-positive tests from self-taken samples in symptomatic individuals. This is especially important in global settings where confirmatory laboratory testing is expensive and unlikely to be readily available.

In conclusion, self-taken swabs for COVID-19 testing offer an accurate alternative to healthcare worker taken swabs for use with RDTs. Our results demonstrate that, with no training, self-taken throat/nasal samples can be used by lay individuals as part of rapid testing programmes for symptomatic adults. Self-testing has the potential to widen access to early diagnosis for COVID-19 in clinical services and outreach settings; allowing access to therapies

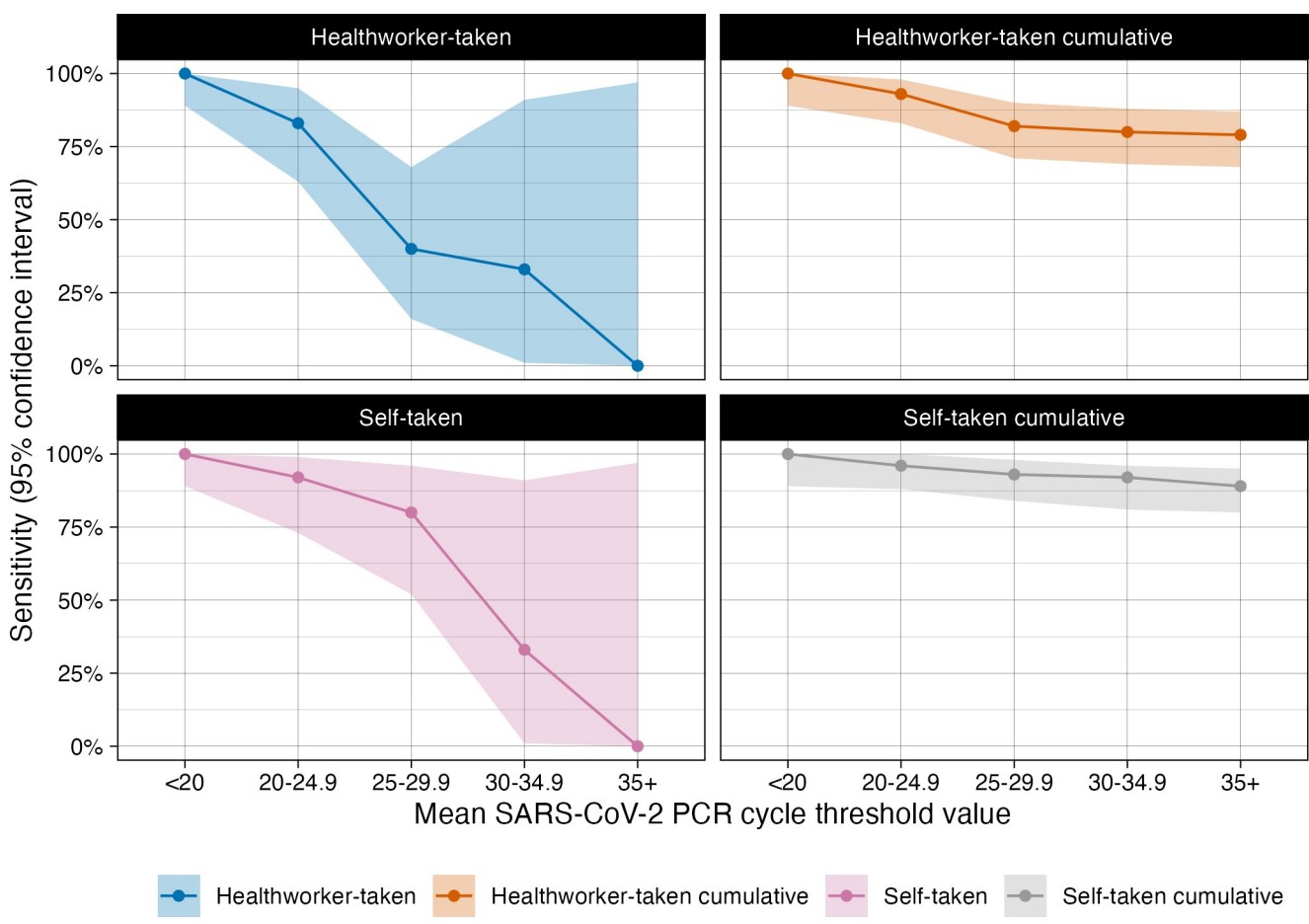

**Fig 2. A graph to show the sensitivity of the Covios®SARS-CoV-2 RDT by mean CT range and cumulative sensitivities.**

in the early stage of the illness. This is especially important where the lack of trained healthcare workers restricts access to testing.

## Supporting information

**S1 Appendix.**
(DOCX)

**S1 Table. A table of all PCR positive participants and their characteristics.**
(DOCX)

**S1 File. Covios® leaflet from study.**
(DOCX)

## Acknowledgments

We acknowledge the support of the National Institute for Health Research (NIHR) Clinical Research Network, which supports delivery of the FALCON study. The views expressed in this article are those of the authors and not necessarily those of the NIHR, or the Department of Health and Social Care.

LSTM diagnostics group: Kate Buist, Karina Clerkin, Dr Thomas Edwards, Dr Susan Gould, Caitlin Greenland-Bews, Konstantina Kontogianni, Laryssa Mashenko, Caitlin R Thompson, Jahanara Wardale, Christopher T Williams and Dominic Wooding.

Lead author LSTM Diagnostics Group: Thomas Edwards, Thomas.edwards@lstmed.ac.uk

Condor steering group: Dr A. Joy Allen, Dr Julian Braybrook, Professor Peter Buckle, Professor Paul Dark, Dr Kerrie Davis, Professor Adam Gordon, Ms Anna Halstead, Dr Charlotte Harden, Dr Colette Inkson, Ms Naoko Jones, Dr William Jones, Professor Dan Lasserson, Dr Joseph Lee, Dr Clare Lendrem, Dr Andrew Lewington, Mx Mary Logan, Dr Massimo Micocci, Dr Brian Nicholson, Professor Rafael Perera-Salazar, Mr Graham Prestwich, Dr D. Ashley Price, Dr Charles Reynard, Dr Beverley Riley, Professor AJ Simpson (Professor Simpson is an NIHR Senior Investigator), Dr Valerie Tate, Dr Philip Turner, Professor Mark Wilcox, Dr Melody Zhifang.

Lead author CONDOR steering group: Richard Body, Richard.body@manchester.ac.uk

## Author Contributions

**Conceptualization:** Helen R. Savage, Richard Body, Gail Hayward, Luis E. Cuevas, Emily R. Adams.

**Data curation:** Helen R. Savage, Lorna Finch, Rachel L. Watkins, Eloïse Cook, Peter MacPherson.

**Formal analysis:** Helen R. Savage, Luis E. Cuevas.

**Funding acquisition:** Richard Body, Gail Hayward, Emily R. Adams.

**Investigation:** Helen R. Savage, Lorna Finch, Rachel L. Watkins, Luis E. Cuevas, Emily R. Adams.

**Methodology:** Helen R. Savage, Lorna Finch, Luis E. Cuevas, Emily R. Adams.

**Project administration:** Helen R. Savage, Lorna Finch, Richard Body, Rachel L. Watkins, Eloïse Cook, Ana I. Cubas-Atienzar.

**Resources:** Lorna Finch, Richard Body, Rachel L. Watkins, Eloïse Cook.

**Software:** Lorna Finch, Richard Body, Eloïse Cook, Peter MacPherson.

**Supervision:** Luis E. Cuevas, Peter MacPherson, Emily R. Adams.

**Validation:** Helen R. Savage, Luis E. Cuevas, Peter MacPherson.

**Visualization:** Helen R. Savage, Luis E. Cuevas, Peter MacPherson.

**Writing – original draft:** Helen R. Savage, Luis E. Cuevas, Peter MacPherson, Emily R. Adams.

**Writing – review & editing:** Helen R. Savage, Lorna Finch, Richard Body, Rachel L. Watkins, Gail Hayward, Eloïse Cook, Ana I. Cubas-Atienzar, Luis E. Cuevas, Peter MacPherson, Emily R. Adams.

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
