## [Decision Letter · Decision Letter 0]

4 Apr 2022

PONE-D-22-05402A prospective diagnostic evaluation of accuracy of self-taken and healthcare worker-taken swabs for rapid COVID-19 testingPLOS ONE

Dear Dr. Savage,

Thank you for submitting your manuscript to PLOS ONE. After careful consideration, we feel that it has merit but does not fully meet PLOS ONE’s publication criteria as it currently stands. Therefore, we invite you to submit a revised version of the manuscript that addresses the points raised during the review process.

This is an interesting study which provides data that are informative for practical use. As two of the reviewers indicate, there should be more details provided on the data to really appreciate this work well.

We look forward to receiving your revised manuscript.

Kind regards,

Sylvia Maria Bruisten, Ph.D

Academic Editor

PLOS ONE

Journal Requirements:

[ERA contributed to this study design and analysis in her role as PhD supervisor to HRS at LSTM, she is also Director of Epidemics and NTDs at Mologic Ltd a UK diagnostics company who provided the RDTs for this study under joint Wellcome funding.] 

3. One of the noted authors is a group or consortium [LSTM Diagnostics group, CONDOR steering group]. In addition to naming the author group, please list the individual authors and affiliations within this group in the acknowledgments section of your manuscript. Please also indicate clearly a lead author for this group along with a contact email address.

Additional Editor Comments:

This is an interesting study which provides data that are informative for practical use. As two of the reviewers indicate there should be more details provided to really appreciate this work well.

Reviewers' comments:

Reviewer's Responses to Questions

**Comments to the Author**

1. Is the manuscript technically sound, and do the data support the conclusions?

Reviewer #1: Partly

Reviewer #2: Partly

Reviewer #3: Partly

2. Has the statistical analysis been performed appropriately and rigorously? 

Reviewer #1: Yes

Reviewer #2: Yes

Reviewer #3: No

3. Have the authors made all data underlying the findings in their manuscript fully available?

Reviewer #1: Yes

Reviewer #2: No

Reviewer #3: No

4. Is the manuscript presented in an intelligible fashion and written in standard English?

Reviewer #1: Yes

Reviewer #2: Yes

Reviewer #3: Yes

5. Review Comments to the Author

Reviewer #1: The paper describes a well conducted study comparing self-taken and HCW taken throat/nasal swabs to perform RDT for SARS-CoV-2. The authors show that self-taken swabs had a higher sensitivity, when compared to RT-PCR and that the NPV of self-taken swabs RDTs was significantly higher than the NPV of HCW-taken RDTs.

Some minor issues:

RT-PCR is used as gold standard, which should be mentioned in the abstract and methods.

Interpretation:

Statements are made on “convenience” and “reduced risk of transmitting infections”, both of which may be true, but are not subject of this study.

As limitation of the study the non-randomized sampling order is mentioned. Reference 15 is cited in line 246 to illustrate that repeated sampling from one nostril does not impact RT-PCR sensitivity or CT values. However this paper shows that this holds for nasal mid-turbinate specimens, while it seems that anterior nasal specimens are used in this study, I think this should be mentioned in the discussion.

Reviewer #2: The manuscripts describes the comparison of self-taken and healthcare worker -taken throat swabs in rapid diagnostic tests (RTDs). The authors show the unexpected result of a higher sensitivity in self taken swab RTDs compared to healthcare taken swabs. The final conclusion of the paper is that RTDs offer substantial individual benefits.

Major comments

- In the methods (line 128) there is a multiplex SARS PCR described, but no data is shown on theses different genes, are there any discrepancies shown? I assume that all analyses are done on the spike protein, but this is not clear in the methods.

- Why is there not chosen to include an extra IC for a household gene (like beta-actin/ beta-globin), this might give some additional data on the quality of the swabs.

- Table 4. The difference in sensitivity is highly influenced by the group of samples with a PCR range 25-29.9, this is not mentioned or discussed. This is probably due to the small sample size in combination with Ct value, and possibly not to the swab method/quality. Are there any control experiments done to support that the turning point of the test indeed is in this range of Ct value.

Minor comment

- Abstract: in my opinion the abstract does not really invite to read the rest of the paper, background is only the method and hypothesis, and the findings is almost only numbers.

- Table 1: Ethnicity, vaccination status and the different symptoms are nicely reported, but there is no further link to the data.

- Table 3/Line 120: I am not sure what is meant by 'read graded'? In the resuts there is also no discrepancies shown in table 3., so I am wondering what the addiotional value is to the test and to the paper.

Reviewer #3: This is an interesting study. In order to give more clarity about the obtained results more details are required.

1. I in the Methods section the data are described. It seems none of the participants has been examined by a medical doctor to identify the health state. Instead, self-assessment as been used. Is this correct?

Specifically, each participant confirmed to have high temperature, contineous cough or change in smell of taste etc. Has the temperature been measures to confirm high temperature? Has any other examination or measurement been made?

Please adjust/modify the provided information correspondingly to reflect the actual situation.

2. If no medical examination has been performed this has to be clearly stated. This is not ideal because the symptoms mentioned in the paper are not unique to Covid-19. This should be also stated.

If all this is the case it seems two tests have been compared without precise information about the underlying health state of the patient.

3. The authors provide an estimate for the sample size assuming a prevelance of 20%. I assume this value is used for the estimate which results in 308 samples? The problem is a prevalence of 20% refers to the population of the UK. In constrast, the authors test only very specific sub-population which self-evaluates as suffering from covid. Hence, the prevalance is different. Please comment and revise correspondingly.

4. On line 157 it is mentioned that 250 participants have been used while the sample size estimate gave 308. Does this mean the study is by design underpowered?

5. The most crucial point of the study relates to the CT value of the RT-PCR test because it is well known that this has a severe impact (add citation).

The authors report different CT values (which is good) but the results therefor are unclear.

Specifically, the results in Tab 2 seem to contradict the literature showing that the higher CT the fewer the positive results. Please clarify this counter intuitive result.

6. The results about the different CT values seem to be the most interesting ones of the study. The discussion should be extended and the finding are highlighted.

7. I did not find information how the data could be obtained.

6. PLOS authors have the option to publish the peer review history of their article (what does this mean?). If published, this will include your full peer review and any attached files.

Reviewer #1: **Yes: **Mirjam Hermans

Reviewer #2: No

Reviewer #3: No

---

## [Author Response · Author response to Decision Letter 0]

9 May 2022

Dear Editor,

We would like to thank the referees for their constructive comments and recommendations made to this manuscript. We have made the revisions suggested and have prepared an itemized reply to all issues raised on the following pages as requested.

We hope these replies are satisfactory to the reviewers and your office. However, please contact us again if you feel there are issues that were not addressed appropriately.

Below is the updated Financial disclosure and competing interests statement:

Financial Disclosure and competing interests

All authors have completed the ICMJE uniform disclosure form at http://www.icmje.org/disclosure-of-interest/ and declare: This study received funding from the UK Research Council through a PhD scholarship from the MRC Doctoral Training Partnership to HRS. PM is funded by Wellcome (200901/Z/16/Z), Wellcome Trust award. ‘Detecting and Excluding Coronavirus disease 2019 (COVID-19) at the Point of Need’ (220764/Z/20/Z), FALCON C-19 study was funded by National Institute for Health Research (NIHR), Asthma UK and the British Lung Foundation; and is supported by the global alliance for diagnostics (FIND). ERA contributed to this study design and analysis in her role as PhD supervisor to HRS at LSTM, she is also Director of Epidemics and NTDs at Mologic Ltd a UK diagnostics company who provided the RDTs for this study under joint Wellcome funding. This does not alter our adherence to PLOS ONE policies on sharing data and materials. For the purpose of open access, the author has applied a CC BY public copyright licence to any Author Accepted Manuscript version arising from this submission. The funders had no role in study design, data collection and analysis, decision to publish, or preparation of the manuscript.

3. One of the noted authors is a group or consortium [LSTM Diagnostics group, CONDOR steering group]. In addition to naming the author group, please list the individual authors and affiliations within this group in the acknowledgments section of your manuscript. Please also indicate clearly a lead author for this group along with a contact email address.

The individual authors alongside their affiliations have been listed in the Acknowledgements section from line 294.

Reviewer #1: The paper describes a well conducted study comparing self-taken and HCW taken throat/nasal swabs to perform RDT for SARS-CoV-2. The authors show that self-taken swabs had a higher sensitivity, when compared to RT-PCR and that the NPV of self-taken swabs RDTs was significantly higher than the NPV of HCW-taken RDTs.

Some minor issues:

RT-PCR is used as gold standard, which should be mentioned in the abstract and methods.

We have updated the abstract and the main methodology to make sure this is clear: 

Line 41 in abstract 

RDT results were compared to RT-PCR as the reference standard to calculate sensitivity and specificity.

Line 136 in the Methods

RT-PCR was used as the reference standard test in this study for comparison of the RDT results.

Interpretation:

Statements are made on “convenience” and “reduced risk of transmitting infections”, both of which may be true, but are not subject of this study.

The abstract interpretation has been rewritten to focus on the study outcomes, line 54:

In conclusion, self-taken swabs for COVID-19 testing offer an accurate alternative to healthcare worker taken swabs for use with RDTs.. Our results demonstrate that, with no training, self-taken throat/nasal samples can be used by lay individuals as part of rapid testing programmes for symptomatic adults. This is especially important where the lack of trained healthcare workers restricts access to testing.

In the discussion line 243 the language has been changed to reflect this:

Using self-sampling for testing could reduce the workload of health workers and increase the ability of services to test patients in both clinical and research settings where trained workers are not available. 

The final paragraph has been rewritten to focus on the study outcomes line 282:

In conclusion, self-taken swabs for COVID-19 testing offer an accurate alternative to healthcare worker taken swabs for use with RDTs. Our results demonstrate that, with no training, self-taken throat/nasal samples can be used by lay individuals as part of rapid testing programmes for symptomatic adults. Self-testing has the potential to widen access to early diagnosis for COVID-19 in clinical services and outreach settings; allowing access to therapies in the early stage of the illness. This is especially important where the lack of trained healthcare workers restricts access to testing. 

As limitation of the study the non-randomized sampling order is mentioned. Reference 15 is cited in line 246 to illustrate that repeated sampling from one nostril does not impact RT-PCR sensitivity or CT values. However this paper shows that this holds for nasal mid-turbinate specimens, while it seems that anterior nasal specimens are used in this study, I think this should be mentioned in the discussion.

Paragraph commencing line 256 rewritten to reflect this:

The limitations of this study are that the sampling order was not randomised as the HCW swab for RDT was taken after the swab for the RT-PCR, and only from one nostril; this may lower sensitivity, or participants may experience sampling fatigue. However, previous studies have shown that repeated sampling from one nostril using nasal mid-turbinate specimens does not impact RT-PCR sensitivity or CT values so, although we used throat/nasal sampling, this repeated sampling may not be the reason for lower sensitivity. 15 Participation was voluntary so people who were less confident to take their own sample may not have agreed to take part; it is also likely participants may have done previous COVID-19 tests, and so have experience of self-sampling.

Reviewer #2: The manuscripts describes the comparison of self-taken and healthcare worker -taken throat swabs in rapid diagnostic tests (RTDs). The authors show the unexpected result of a higher sensitivity in self taken swab RTDs compared to healthcare taken swabs. The final conclusion of the paper is that RTDs offer substantial individual benefits.

Major comments

- In the methods (line 128) there is a multiplex SARS PCR described, but no data is shown on theses different genes, are there any discrepancies shown? I assume that all analyses are done on the spike protein, but this is not clear in the methods.

A mean CT value was calculated from the genes tested to give a single CT value for analysis. This has been updated in the sample size and statistical analysis section and also stated in the results with reference to the full data set provided as a supplementary table:

Line 145

For each RT-PCR result with three target genes, the mean of the three CT values was taken to give a single RT-PCR CT result for each sample; RT-PCR results with mean cycle threshold (CT) values <40 were considered positive and CT values ≥40 were graded as negative.

Line 218

Sensitivity of the RDTs varied by mean CT values (Table 4 and Figure 2); a full table of mean and CT values from each gene tested is available in the supplementary data.

- Why is there not chosen to include an extra IC for a household gene (like beta-actin/ beta-globin), this might give some additional data on the quality of the swabs.

The PCR used was a WHO approved PCR and therefore we were not able to make alterations to it. We believe this is a really interesting idea and we believe would make a complementary piece of follow-up using an in-house PCR however was outside the scope of this study. 

- Table 4. The difference in sensitivity is highly influenced by the group of samples with a PCR range 25-29.9, this is not mentioned or discussed. This is probably due to the small sample size in combination with Ct value, and possibly not to the swab method/quality. Are there any control experiments done to support that the turning point of the test indeed is in this range of Ct value.

Line 264 an additional limitation was added to discuss the influence this small group of results has had on the overall sensitivity:

The majority of participants with discrepant results between self-taken and HCW taken RDT results had a mean CT of between 25-30 on PCR (six of ten). There was only a small number of participants with a CT value within this range however this group makes a large difference to the sensitivity of the HCW taken RDT tests. This group was too small to analyse further statistically; it is noted similar results were noted in the trial by García-Fiñana et al 4 in the group with a CT greater than 24.4. In future studies evaluating RDTs for SARS-CoV-2 this may be an important subset to review as this may influence test performance.

Minor comment

- Abstract: in my opinion the abstract does not really invite to read the rest of the paper, background is only the method and hypothesis, and the findings is almost only numbers.

Thank you for this feedback, we have revised the abstract at the start of the work in line with the comments of all the reviewers.

- Table 1: Ethnicity, vaccination status and the different symptoms are nicely reported, but there is no further link to the data.

On line 177 a link to the complete dataset within the supplementary files is provided.

A full data table of all participant characteristics and results is available in the supplementary data.

- Table 3/Line 120: I am not sure what is meant by 'read graded'? In the results there is also no discrepancies shown in table 3., so I am wondering what the additional value is to the test and to the paper.

The Covios test used has a visual grading card for the results line to give a numerical result. We reported the results as per the manufacturers guidance and for completion included these results in the manuscript. We have updated the description of the reading of the tests, line 125:

RDTs were run according to the manufacturer’s instructions; this includes grading the result line from 0-10 on the RDT using a visual reference card. Each test was read by two trained researchers, if there was a disagreement between the two readers, a third reader was requested.

Table 3 has been updated to focus on the positive/negative results with the visual graded result in brackets as for the CT value.

Participant RT-PCR result (mean Ct value) Self-taken RDT result (Reader 1/Reader 2) HCW-taken RDT result (Reader 1/Reader 2)

26 Positive (29.03) Positive (4/4) Negative (0/0)

38 Negative (>40) Negative (0/0) Positive (2/2)

106 Positive (21.68) Positive (3/3) Negative (0/0)

131 Positive (27.99) Positive (3/3) Negative (0/0)

140 Positive (25.44) Positive (3/3) Negative (0/0)

169 Positive (28.09) Positive (2/2) Negative (0/0)

189 Positive (20.35) Positive (8/8) Negative (0/0)

195 Positive (22.96) Positive (3/3) Negative (0/0)

231 Positive (27.33) Positive (6/6) Negative (0/0)

249 Positive (27.86) Positive (5/5) Negative (0/0)

Reviewer #3: This is an interesting study. In order to give more clarity about the obtained results more details are required.

1. I in the Methods section the data are described. It seems none of the participants has been examined by a medical doctor to identify the health state. Instead, self-assessment as been used. Is this correct?

Specifically, each participant confirmed to have high temperature, continuous cough or change in smell of taste etc. Has the temperature been measures to confirm high temperature? Has any other examination or measurement been made?

Please adjust/modify the provided information correspondingly to reflect the actual situation.

Line 102 in the Methods section has been clarified:

Demographic and self-reported symptom data were recorded electronically; a list of symptoms from the FIND alliance and participants could also list additional symptoms they felt were relevant. 

Line 174 has also been clarified in the Results section:

The most common self-reported symptoms by participants were… 

2. If no medical examination has been performed this has to be clearly stated. This is not ideal because the symptoms mentioned in the paper are not unique to Covid-19. This should be also stated.

We have clarified the self-reporting of symptoms as above in response to the first comments above.

If all this is the case it seems two tests have been compared without precise information about the underlying health state of the patient.

Using a prospective evaluation we aimed to account for the fact that the majority of symptomatic participants were unlikely to have current SARS-CoV-2 infection therefore we calculated a sample size based on 20% prevalence of SARS-CoV-2 in the tested population in this setting in the UK. Using RT-PCR positivity as the gold standard to assess for presence or absence of SARS-CoV-2 infection allowed us to compare the two RDTs taken concurrently.

3. The authors provide an estimate for the sample size assuming a prevalence of 20%. I assume this value is used for the estimate which results in 308 samples? The problem is a prevalence of 20% refers to the population of the UK. In contrast, the authors test only very specific sub-population which self-evaluates as suffering from covid. Hence, the prevalence is different. Please comment and revise correspondingly.

This prevalence was taken from the positivity testing rate for SARS-CoV-2 in Liverpool where the study was to take place in the week commencing 21st January 2021 when the study size was calculated. This was the prevalence in people self-identifying COVID-19 and presenting for testing in Liverpool U.K. We conducted the study within a PCR testing site, so this was the prevalence in the population we were studying. 

We have updated the text at line 138 to clarify this point:

Sample size was calculated using an alpha of 0.05, anticipated prevalence of 20% (based on the positivity rate of PCR tests of individuals presenting for testing within Liverpool in the week commencing 21st January 2021, calculated by Public Health England 10), minimum test sensitivity of 80%, specificity 99%, and precision interval of 10%.

4. On line 157 it is mentioned that 250 participants have been used while the sample size estimate gave 308. Does this mean the study is by design underpowered?

Recruitment was halted early as high number of participants tested positive by RDT meaning prevalence was higher than in the calculated sample size and it was felt no further information would be gained by recruiting further participants.

5. The most crucial point of the study relates to the CT value of the RT-PCR test because it is well known that this has a severe impact (add citation).

The authors report different CT values (which is good) but the results therefor are unclear.

Specifically, the results in Tab 2 seem to contradict the literature showing that the higher CT the fewer the positive results. Please clarify this counter intuitive result.

For PCR methodology the more amplification cycles needed to identify the virus mean that less virus is present initially, therefore this is consistent with a lower positivity rate of RDT tests at higher CT values (as more cycles were needed to get a positive result). Within this test therefore this is a result that is consistent and found in many studies, including within the population in Liverpool.

6. The results about the different CT values seem to be the most interesting ones of the study. The discussion should be extended and the finding are highlighted.

We have extended the discussion around CT values and positivity rates as described above in response to the comments above from reviewer 2 and also this feedback and the comments above:

Line 264 - The majority of participants with discrepant results between self-taken and HCW taken RDT results had a mean CT of between 25-30 on PCR (six of ten). There was only a small number of participants with a CT value within this range however this group makes a large difference to the sensitivity of the HCW taken RDT tests. This group was too small to analyse further statistically; it is noted similar results were noted in the trial by García-Fiñana et al 4 in the group with a CT greater than 24.4. In future studies evaluating RDTs for SARS-CoV-2 this may be an important subset to review as this may influence perceived test performance.

7. I did not find information how the data could be obtained.

Clarification added to Line 177 that a full data table is available in the supplementary material:

A full data table of all participant characteristics and results is available in the supplementary data.

---

## [Decision Letter · Decision Letter 1]

7 Jun 2022

PONE-D-22-05402R1A prospective diagnostic evaluation of accuracy of self-taken and healthcare worker-taken swabs for rapid COVID-19 testingPLOS ONE

Dear Dr. Savage,

Thank you for submitting your manuscript to PLOS ONE. After careful consideration, we feel that it has merit but does not fully meet PLOS ONE’s publication criteria as it currently stands. Therefore, we invite you to submit a revised version of the manuscript that addresses the points raised during the review process.

 Since only one of the previous reviwers was available to look at the rebuttal I have now also looked at it as reviewer number 4. There is still a comment on the lack of (internal) controls and the small group of samples with relatively high Ct values, which have a large influence on the results. Please clarify if this has an impact on the main message of your paper.

We look forward to receiving your revised manuscript.

Kind regards,

Sylvia Maria Bruisten, Ph.D

Academic Editor

PLOS ONE

Journal Requirements:

Additional Editor Comments:

Since there was only one of the previous reviewers available to look at the rebuttal I have now also looked at it as reviewer number 4.

Reviewer 2 still has a problem with the comment that there is a large influence of the group of samples with Ct values ranging between 25 and 30 and that this is a small group. If possible you should further discuss why you think that the data are still sound. Please make more clear that this will not influence the message of the paper.

Reviewers' comments:

Reviewer's Responses to Questions

**Comments to the Author**

1. If the authors have adequately addressed your comments raised in a previous round of review and you feel that this manuscript is now acceptable for publication, you may indicate that here to bypass the “Comments to the Author” section, enter your conflict of interest statement in the “Confidential to Editor” section, and submit your "Accept" recommendation.

Reviewer #2: (No Response)

Reviewer #4: All comments have been addressed

2. Is the manuscript technically sound, and do the data support the conclusions?

Reviewer #2: Partly

Reviewer #4: Yes

3. Has the statistical analysis been performed appropriately and rigorously? 

Reviewer #2: I Don't Know

Reviewer #4: Yes

4. Have the authors made all data underlying the findings in their manuscript fully available?

Reviewer #2: Yes

Reviewer #4: Yes

5. Is the manuscript presented in an intelligible fashion and written in standard English?

Reviewer #2: Yes

Reviewer #4: Yes

6. Review Comments to the Author

Reviewer #2: In my opinion the comments are adressed only minor, and despite the conlusions are discussed in a better way, the paper still lacks the proper controls/control experiments supporting the differences shown driven by the group with PCR positivity 25-29.

Reviewer #4: The authors did a good job to answer all points made by the reviewers. It is an interesting study that deserves to be published.

7. PLOS authors have the option to publish the peer review history of their article (what does this mean?). If published, this will include your full peer review and any attached files.

Reviewer #2: No

Reviewer #4: No

---

## [Author Response · Author response to Decision Letter 1]

10 Jun 2022

Dear Editor,

We would like to thank the editor for their constructive comments and recommendations made to this manuscript. We have reviewed the feedback from yourself and Reviewer 2 and have made the revisions suggested and have prepared an itemized reply to all issues raised on the following pages as requested.

We have especially focused on the comments and recommendations from Reviewer 2 and have reviewed our data to address their concerns. We have ensured that the subset of participants does not change the outcome or message of the paper and made this much clearer in the limitations section of our work. We hope that the addition of this information will be satisfactory and have also included their suggestions for further work which we are sadly unable to carry out as we do not have the scope or resources to undertake additional experiments at this time.

We hope these replies are satisfactory to the reviewers and your office.

We look forward to your decision.

Best regards,

Helen Savage

On behalf of all authors.

PONE-D-22-05402R1

A prospective diagnostic evaluation of accuracy of self-taken and healthcare worker-taken swabs for rapid COVID-19 testing

Response to comments from Editor

Since only one of the previous reviewers was available to look at the rebuttal I have now also looked at it as reviewer number 4. There is still a comment on the lack of (internal) controls and the small group of samples with relatively high Ct values, which have a large influence on the results. Please clarify if this has an impact on the main message of your paper.

Reviewer 2 still has a problem with the comment that there is a large influence of the group of samples with Ct values ranging between 25 and 30 and that this is a small group. If possible you should further discuss why you think that the data are still sound. Please make more clear that this will not influence the message of the paper.

Thank you for these helpful comments, and we will respond to the comments from the Editor and from Reviewer to together here. 

We have carefully rereviewed the data in the manuscript and revised the wording of the text in the Discussion section to make clear that the subset of data would not change the overall conclusion of the paper if excluded and further work that could be undertaken to characterise this group.

The revised text (Line 265 – 279) now reads:

The majority of participants with discrepant results between self-taken and HCW taken RDT results had a mean CT of between 25-30 on PCR (six of ten). Although there was only a small number of participants with a CT value in this range, these values are influential in estimates of the sensitivity of the HCW taken RDT tests. Because of the small numbers, we did not undertake further formal statistical analysis; however when participants with CT values within this range were excluded, our main conclusion that self-taken swabs were non-inferior to healthcare worker taken swabs did not change (sensitivity with CT 25-29.9 excluded: self taken RDT 91.7% [55/60]; HCW taken 88.3% [53/55]). Similar results were noted in the trial by García-Fiñana et al4 who also saw a decrease in sensitivity of RDTs compared to PCR in samples with a CT of greater than 24.4. Future research could compare self-taken and healthcare worker-taken samples by PCR including housekeeper genes to see if the quality of sampling varies and may affect CT values, particularly for samples with CT values over 25, as lower sampling quality could result in lower CT values and therefore a perceived poorer performance in this range.

Response to Comments from Reviewer 2

In my opinion the comments are addressed only minor, and despite the conclusions are discussed in a better way, the paper still lacks the proper controls/control experiments supporting the differences shown driven by the group with PCR positivity 25-29.

Thank you for the suggestion. Please additionally see the Response to the Editor’s comments above where we have aimed to make it clear that this small group does not affect the conclusions of the paper. The PCR used was a WHO approved PCR and therefore we were not able to make alterations to it but we also agree this is an interesting idea and we believe would make a complementary piece of follow-up work using an in-house PCR and have therefore included this within the limitations as a piece of additional experimentation that would be beneficial as future research.

---

## [Editor Report · Decision Letter 2]

16 Jun 2022

A prospective diagnostic evaluation of accuracy of self-taken and healthcare worker-taken swabs for rapid COVID-19 testing

PONE-D-22-05402R2

Dear Dr. Savage,

We’re pleased to inform you that your manuscript has been judged scientifically suitable for publication and will be formally accepted for publication once it meets all outstanding technical requirements.

Kind regards,

Sylvia Maria Bruisten, Ph.D

Academic Editor

PLOS ONE

Additional Editor Comments (optional):

All points were addressed, also the last one, specifying that the the small group of samples with elevated Ct values did not alter the conclusions.
---

## [Editor Report · Acceptance letter]

22 Jun 2022

PONE-D-22-05402R2 

A prospective diagnostic evaluation of accuracy of self-taken and healthcare worker-taken swabs for rapid COVID-19 testing 

Dear Dr. Savage:

I'm pleased to inform you that your manuscript has been deemed suitable for publication in PLOS ONE. Congratulations! Your manuscript is now with our production department. 

Kind regards, 

on behalf of

Dr. Sylvia Maria Bruisten 

Academic Editor

PLOS ONE